# Pulmonary fibroblasts activated by the addition of TNF-α and IL-4 enhance lymphangiogenic capacity and ameliorate lung fibrosis in an allogeneic rat model

Yuimi Matsuoka[1,2*], Yuuki Shimizu[1], Koji Sakamoto[3], Makoto Matsuyama[4], Toyoaki Murohara[1], Takahiro Iwamiya[2,5*]

**1** Department of Cardiology, Nagoya University Graduate School of Medicine, Nagoya, Japan, **2** Research & Development Department, LYMPHOGENiX Ltd, Covent Garden, London, England, United Kingdom, **3** Department of Respiratory Medicine, Nagoya University Graduate School of Medicine, Nagoya, Japan, **4** Department of Otorhinolaryngology, Juntendo University Faculty of Medicine, **5** Institute for Advanced Biosciences, Keio University, Tsuruoka, Yamagata, Japan

* yuimi@lymphogenix.com (YM), takahiro@lymphogenix.com (TI)

## Abstract

### Background

Pulmonary fibrosis remains a major clinical challenge with limited treatment options. Recent studies have suggested that fibroblasts, when stimulated by specific cytokines, may acquire lymphangiogenic and antifibrotic properties contributing to tissue repair.

### Methods

Human and rat pulmonary fibroblasts (PFs) were stimulated with TNF-α and IL-4 to induce lymphangiogenic and antifibrotic characteristics. In vitro analyses assessed gene expression, cytokine secretion, tube formation capacity, and immunogenicity. Therapeutic efficacy was evaluated in a rat model of bleomycin-induced pulmonary fibrosis following allogeneic PF transplantation.

### Results

Cytokine-stimulated PFs exhibited upregulation of *ADM* and *VEGFC*, enhanced tube formation capacity, and minimal expression of immunogenic markers. In vivo, allogeneic PF transplantation significantly reduced fibrotic lesion and plasma SP-D levels compared to controls. Gene expression analyses demonstrated downregulation of fibrosis-associated markers after treatment.

### Conclusion

Cytokine-stimulated pulmonary fibroblasts may serve as a novel cell source for antifibrotic therapy by modulating lymphangiogenesis and tissue remodeling, providing

**Data availability statement:** All relevant data are within the manuscript and its Supporting Information files.

**Funding:** This research was supported through a collaborative research agreement with LYMPHOGENiX Ltd. LYMPHOGENiX Ltd. provided support in the form of salaries for authors YM and TI, and covered research-related expenses including personnel costs. All aspects of the study, including study design, experimental execution, data collection, data analysis, and interpretation of results, were conducted independently at Nagoya University, where the study was performed. LYMPHOGENiX Ltd. did not have any additional role in the study design, data collection and analysis, decision to publish, or preparation of the manuscript.

**Competing interests:** I have read the journal's policy and the authors of this manuscript have the following competing interests: YM and TI are employed by/affiliated with LYMPHOGENiX Ltd. This does not alter our adherence to PLOS ONE policies on sharing data and materials. These revisions clarify that LYMPHOGENiX Ltd. provided financial support through a collaborative research agreement, including salaries for authors YM and TI, while the research was conducted independently at Nagoya University. We confirm full adherence to PLOS ONE data and materials sharing policies.

a potential alternative to conventional stem cell-based approaches for fibrotic lung diseases.

## Introduction

Idiopathic pulmonary fibrosis (IPF) and other chronic forms of pulmonary fibrosis are progressive and life-threatening disorders characterized by excessive extracellular matrix (ECM) deposition and architectural distortion of the lung, ultimately leading to irreversible respiratory failure and poor prognosis [1,2]. Despite the availability of antifibrotic agents such as pirfenidone and nintedanib, these therapies only slow the functional decline and fail to reverse established fibrosis [3]. Therefore, novel regenerative approaches are urgently required.

Persistent alveolar epithelial injury activates fibroblasts and promotes their differentiation into myofibroblasts, driving excessive ECM accumulation and fibrosis progression [4,5]. Recently, emerging therapeutic strategies such as lipid nanoparticle (LNP)-mediated mRNA delivery of CAR-T cells targeting activated fibroblasts have shown promising results in preclinical fibrosis models [6], although their clinical application in pulmonary fibrosis remains at an early stage.

Cell-based therapies, particularly mesenchymal stromal cells (MSCs), have attracted considerable attention, demonstrating antifibrotic and immunomodulatory effects in preclinical models of pulmonary fibrosis and early-phase clinical trials [7,8]. However, challenges such as donor variability, immunogenicity, and manufacturing complexity remain significant obstacles to clinical translation [9,10].

Interestingly, our previous studies in cardiac fibrosis models, including both heart failure with reduced ejection fraction (HFrEF) and heart failure with preserved ejection fraction (HFpEF), have demonstrated that fibroblasts, when stimulated with specific cytokines, can acquire reparative properties by secreting lymphangiogenic factors such as *ADM* and *VEGFC*. The promotion of lymphangiogenesis was associated with extracellular matrix remodeling and fibrosis reduction in these cardiac models [11–13]. Based on these findings, we hypothesized that similar fibroblast-mediated lymphangiogenic mechanisms may contribute to fibrosis resolution in pulmonary fibrosis.

In this study, we investigated whether human and rat pulmonary fibroblasts (PFs), when stimulated with TNF-α and IL-4, acquire lymphangiogenic and antifibrotic properties. We evaluated gene expression, tube formation capacity, cytokine secretion, and immunogenicity in vitro, and examined therapeutic efficacy following allogeneic PF transplantation in a rat model of bleomycin-induced pulmonary fibrosis.

## Materials and methods

The authors declare that all supporting data are available within the article and Online Data Supplement. Raw data are available from the corresponding author on request, where reasonable.

Antibodies and reagents for magnetic-activated cell sorting (MACS), fluorescence-activated cell sorting (FACS), staining, and immunohistochemistry

Antibodies and reagents used for FACS are listed in S1 Table.

## Expansion of human PFs and creating human PFs+CKs

Human pulmonary fibroblasts were purchased from PromoCell (Heidelberg, Germany) and cultured with HFDM-1(+) medium (Cell Science & Technology, Osaka, Japan) supplemented with 5% (v/v) Newborn Calf Serum (NBCS) and 1% (v/v) Penicillin-Streptomycin (P/S). After initial expansion (passage 6−8), fibroblasts were incubated with 50 ng/mL of recombinant human TNF-a and 2 ng/mL of recombinant human IL-4 for 72 hr in order to create hPFs+CKs.

## Expansion of rat PFs and creating rat PFs+CKs

Rat pulmonary fibroblasts were isolated from Wistar Kyoto (WKY) rats and primary culture was performed. The growing medium was DMEM Low-Glucose(nacalai tesque, Kyoto, Japan) supplemented with 15% (v/v) Fetal Bovine Serum (FBS) and 1% (v/v) Penicillin-Streptomycin (P/S). Similarly to human pulmonary fibroblasts, rat pulmonary fibroblasts were incubated with 50 ng/mL of recombinant human TNF-a and 2 ng/mL of recombinant human IL-4 for 72 hr in order to create rPF+CKs.

## Immuno staining of cells, staining

Cells were incubated with primary and secondary antibodies (for 30 minutes each). The antibodies used for immunostaining are listed in S1 Table. FACS analyses were performed with a MACSQuant Analyzer following the manufacturer's instructions (Miltenyi Biotec).

## Animal welfare and experimental procedures

All animal care and research procedures were approved by the Institutional Animal Care and Use Committee of KAWASAKI INSTITUTE OF INDUSTRIAL PROMOTION Innovation Center of NanoMedicine (Approval number: A21-003–3). All experiments were conducted in accordance with institutional guidelines and the ARRIVE guidelines. Male BN/CrlCrlj rats (6 weeks old, 130–200 g) were purchased from The Jackson Laboratory Japan (Yokohama, Japan) and housed under standard laboratory conditions (temperature: 22±2°C, humidity: 50±10%, 12-hour light/dark cycle) with free access to food and water. Animals were allowed a 1-week acclimatization period before experimental procedures.

For bleomycin administration and cell transplantation procedures, rats were anesthetized with a triple-combination anesthetic cocktail (5 mL/kg body weight, intraperitoneally) consisting of medetomidine (Domitor, 2 mL), butorphanol (Vetorphale, 2.5 mL), midazolam (2 mL), and physiological saline (18.5 mL) for a total volume of 25 mL. No additional post-operative analgesia was administered as the procedures were minimally invasive and animals showed no signs of distress during recovery.

All rats were administered bleomycin (5 mg/mL, 1 mL/kg; equivalent to 5 mg/kg body weight) by microspray through the trachea after the acclimatization period. The Sham group received an equivalent volume of saline instead of bleomycin. Five days after bleomycin treatment, hPF, hPF+CKs, or saline (control) was administered via tail vein injection.

Animals were monitored daily for body weight, respiratory rate, activity level, and general appearance throughout the experimental period. Humane endpoints were prospectively defined as: (1) body weight loss exceeding 20% of baseline, (2) severe respiratory distress (respiratory rate >150 breaths/min at rest or labored breathing), (3) inability to access food or water, (4) severe lethargy or persistent hunched posture for >24 hours, or (5) any other signs of severe suffering. Animals meeting these criteria would be immediately euthanized; however, no animals reached these endpoints during the study, and all animals survived until the scheduled endpoint.

Two weeks after cell or saline administration, animals were euthanized by exsanguination under deep anesthesia (isoflurane 5% induction followed by maintenance at 3–4% until loss of reflexes). Death was confirmed by absence of heartbeat and corneal reflex. Lung tissue and blood were collected for subsequent analysis.

All efforts were made to minimize animal suffering throughout the study. The bleomycin dose and 2-week observation period were selected based on established protocols to induce reproducible fibrosis while limiting the duration of animal distress.

## Tissue section preparation and staining

Lung sections were fixed with formalin and embedded in paraffin. Collagen fibers were stained with Masson trichrome staining (MT). Stained sections were imaged with BZ-X810(KEYENCE). Quantification of fibrosis was performed using the method described in previous reports.

## In situ hybridization(ISH) of paraffin sections

Double staining for antibody staining of WGA and ISH of Lyve-1 was performed using FFPE prepared in the previous section. First, FFPE sections were deparaffinised using xylene and ethanol, and then Blocking One Hist (nacalai tesque, Kyoto, Japan) was applied for 10 min at room temperature. After washing with D-PBS, primary antibody (Goat Anti- Wheat Germ Agglutinin, Unconjugated Antibody (Vector, CAT.#AS-2024–1)) was reacted at 4 overnight. After washing with D-PBS again, the secondary antibody (Donkey Anti-Goat IgG H&L (Alexa Fluor 488) (abcam, CAT. #150129) was reacted at room temperature for 40 min. The process then proceeded to the ISH process.

The ISH procedure was carried out according to the manual supplied with the ViewRNA "Tissue Fluorescence Assay". The heat treatment time for these samples was set to 10 min and the protease digestion time to 20 min. The wavelength of Lyve-1 was set to 647 nm. Counterstaining was also performed with DAPI contained in the kit. Finally, autofluorescence was suppressed using the ReadyProbes™ Tissue Autofluorescence Quenching Kit (Invitrogen, CAT. #R37630) and mounted.

Sections were imaged on a SpinSR10 (Evident, Tokyo, Japan) a common instrument of the Nagoya University Graduate School of Medicine.

## RNA extraction and quantitative PCR

Total RNA was extracted from minced rat lung tissue using the acid guanidinium thiocyanate-phenol-chloroform (AGPC) method with Sepazol-RNA I Super G (Nacalai Tesque, 09379−84). Briefly, lung tissue was minced with a razor blade and homogenized in 500 µL of Sepazol using a Biomasher V homogenizer (Nippi, 891390). The subsequent steps were carried out according to the manufacturer's instructions. The resulting RNA pellet was dissolved in 100 µL of sterile, RNase-free water. RNA concentration and purity were assessed using a NanoDrop spectrophotometer, and samples were stored at −80 °C until further use.

Complementary DNA (cDNA) was synthesized from total RNA using the ReverTra Ace qPCR RT Master Mix with gDNA Remover (Toyobo, RSQ-301), following the manufacturer's instructions. Quantitative PCR (qPCR) was performed using PowerTrack SYBR Green Master Mix (Thermo Fisher Scientific, A46109) on a ViiA 7 Real-Time PCR System (Thermo Fisher Scientific).

Thermal cycling conditions were as follows: an initial denaturation step at 95 °C for 2 minutes, followed by 40 cycles of denaturation at 95 °C for 5 seconds and annealing/extension at 60 °C for 30 seconds. Melting curve analysis was performed at the end of each run to confirm amplification specificity.

Target genes related to fibrosis and inflammation included *Acta2, Col1a1, Tnfa, Tgfb, Il6,* and *Ccl2.* Lymphangiogenesis-related genes *Lyve1, Prox1*, and *Flt4* were also analyzed. Two housekeeping genes, *B2m* and *Gusb*, were used as internal controls. Gene expression was quantified using standard curves and normalized using the Pfaffl method, incorporating both internal control genes for inter-sample normalization.

## Tube formation and immune-staining

Lung microlymphatic endothelial cells (HMLECs) purchased from Lonza (Walkersville, MD, USA) were expanded and cultured in EBM-2–80% confluence. The cells were then collected and direct co-cultured with hPF or hPF + CKs, for 3 days. At this time, HMLECs were mixed at $8.32 \times 10^3$ cells/well and hPF or hPF + CKs at $7.49 \times 10^3$ cells/well, and EGM-2 was used as the medium. During direct co-culture, the medium was not changed.

After direct co-culture, the cells were fixed with 4% Paraformaldehyde Phosphate Buffer Solution. 0.1% Triton-X was then added for membrane permeabilisation and the cells were allowed to stand at room temperature for 15 min. The cells then proceeded to the immunostaining process.

To perform the primary antibody reaction, Anti-VE Cadherin Rabbit polyclonal Antibody (abcam,CAT.#ab33168) and Anti-Vimentin Mouse polyclonal Antibody (abcam,CAT.#ab20346,) were added and reacted at 4°C overnight.

After washing with D-PBS, the reaction proceeded to a secondary antibody reaction with Goat Anti-Rabbit IgG H&L (AlexaFluor488) (abcam,CAT.#150077) and Goat Anti- Mouse IgG H&L (Alexa Fluor 647) (abcam, CAT. #150115) were added and the cells were light-shielded and reacted at 4°C for 3 hours.

To perform nuclear staining, cells were washed with D-PBS, then Hoechst33258 solution (Dojindo, CAT.#343–17961) was added and reacted for 15 min at room temperature.

Images were then taken on an InCell Analyzer 2200 (GE Healthcare) and the brightness of each fluorescence was adjusted using Image J software.

## ELISA assay

Concentrations of pulmonary surfactant protein D (SP-D) and fibrinogen in rat plasma samples were determined using ELISA kits (Rat Fibrinogen ELISA Kit (abcam, ab108846), Rat/Mouse SP-D Kit (Yamasa Shoyu, 80072).

## Data analysis

Data are presented as mean ± standard deviation (SD) or mean ± standard error (SE). Statistical analyses were performed using [GraphPad Prism version10.3.0].

For comparisons between two groups, unpaired Student's t-test was used when. data met the assumptions of normality and equal variance. When the assumption of equal variance was violated, Welch's t-test was applied.

For comparisons among three or more groups, one-way analysis of variance (one-way ANOVA) was used when data met the assumptions of normality and equal variance, followed by [Tukey's/ Dunnett's/ Bonferroni's] multiple comparison test to identify specific group differences. When data did not meet the assumption of normal distribution, the non-parametric Kruskal-Wallis test was used, followed by Dunn's multiple comparisons test.

Statistical significance was set at $P < 0.05$. In all figures, *$P < 0.05$, **$P < 0.01$, and ***$P < 0.001$ denote significant differences between groups as indicated.

## Results

### TNF-α and IL-4 induce CD90⁺VCAM-1⁺ phenotype and lymphangiogenic potential in human pulmonary fibroblasts

Human pulmonary fibroblasts (hPFs) were initially expanded on culture dishes or flasks. These cells exhibited a typical fibroblastic morphology, characterized by a flat and spindle-shaped appearance. Upon stimulation with TNF-α and IL-4 (hPFs + CKs), the cells exhibited further elongation and morphological changes, resembling those observed in human cardiac fibroblasts [13] (Fig 1A).

Flow cytometry (FCM) analysis was performed to assess the surface expression of CD90 and VCAM-1 in hPFs and hPFs + CKs, In hPFs + CKs, VCAM-1 expression was elevated, and the proportion of double-positive (CD90⁺VCAM-1⁺) cells increased from 15.56% to 28.45% (Fig 1B).

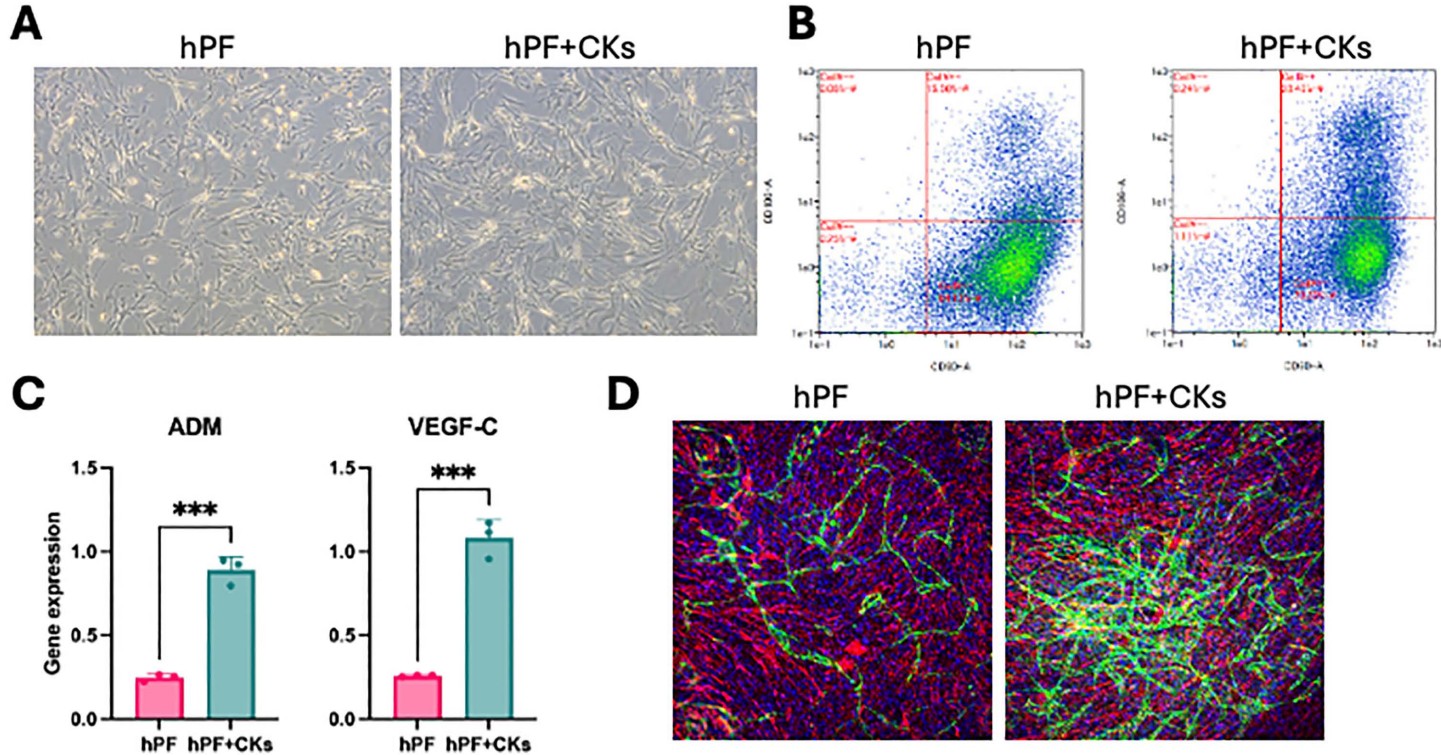

**Fig 1. Generation of hPFs and acquisition of lymphangiogenic potential. (A)** Representative phase-contrast images of hPFs without stimulation (left) and hPFs stimulated with TNF-α and IL-4 (hPF + CKs, right). **(B)** Flow cytometry of CD90 and VCAM-1 expression on the cell surface of hPFs and hPF + CKs, with CD90 and VCAM-1 plotted on the x- and y-axes, respectively.The percentages of cells in each quadrant are as follows: · hPFs: CD90⁺VCAM-1⁺ = 15.56%, CD90⁺VCAM-1⁻ = 84.13%, CD90⁻VCAM-1⁺ = 0.06%, CD90⁻VCAM-1⁻ = 0.25% · hPF + CKs: CD90⁺VCAM-1⁺ = 28.45%, CD90⁺VCAM-1⁻ = 70.20%, CD90⁻VCAM-1⁺ = 0.24%, CD90⁻VCAM-1⁻ = 1.11% **(C)** mRNA expression levels of *ADM* and *VEGFC* in hPFs and hPF + CKs determined by RT-qPCR. Data are presented as mean ± SEM. ***$P < 0.001$, ****$P < 0.0001$ by unpaired t-test. **(D)** Tube formation assay using co-culture with HMVECs. Immunofluorescence images: Vimentin (red), VE-Cadherin (green), and Hoechst 33258 (blue).

To assess potential differences in lymphangiogenic capacity, gene expression levels of adrenomedullin (*ADM*) and *VEGFC* were measured by RT-qPCR. Both genes were significantly upregulated in hPFs + CKs compared to untreated hPFs (Fig 1C). This finding was corroborated by an in vitro lymphangiogenesis assay, in which hPFs + CKs formed multiple, elongated, and structurally robust lymphatic vessel-like structures, compared to hPFs (Fig 1D).

These results suggest that TNF-α and IL-4 stimulation endows hPFs with lymphangiogenic properties, comparable to those observed in human cardiac fibroblasts [13].

### Human pulmonary fibroblasts and cytokine-stimulated counterparts exhibit low immunogenicity

The immunogenicity of hPFs and cytokine-stimulated hPFs (hPFs + CKs) was assessed by flow cytometry. Both cell types consistently expressed HLA Class I molecules, which present endogenous peptides and are essential for self-recognition and immune tolerance. In contrast, HLA Class II molecules, typically expressed on professional antigen-presenting cells and responsible for presenting exogenous peptides, were not detected. Similarly, the co-stimulatory molecules CD80 and CD86, which are critical for T cell activation, were also absent. These findings indicate that both hPFs and hPFs + CKs exhibit low immunogenicity, supporting their potential use as allogeneic cell sources in clinical applications (Fig 2).

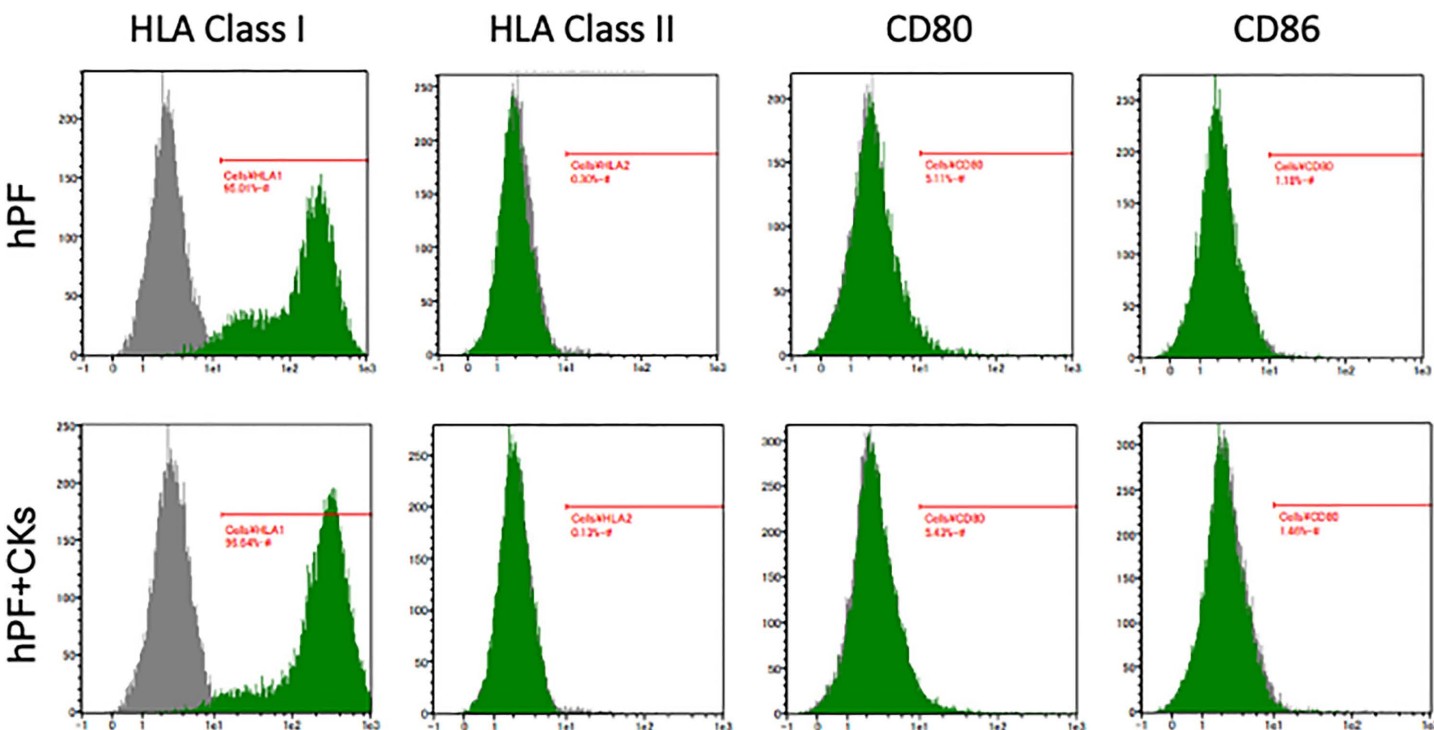

**Fig 2. Immunogenicity of adult human pulmonary fibroblasts.** Flow cytometry of immune-related markers HLA class I, HLA class II, CD80, and CD86 in hPFs and hPF+CKs.

## Assessment of allogeneic fibroblast therapy in a rat model of pulmonary fibrosis

To explore the therapeutic applicability of allogeneic fibroblast transplantation for pulmonary fibrosis, we employed a donor-recipient rat model using genetically distinct inbred strains. Pulmonary fibrosis was induced in BN/CrlCrlj (BN) rats by intratracheal administration of bleomycin (BLM) via microspray. Five days after BLM administration, donor pulmonary fibroblasts (rPFs) isolated from WKY/NCrlCrlj (WKY) rats, either untreated or pre-stimulated with TNF-α and IL-4 (rPF+CKs) were intravenously transplanted into the recipient BN rats. A saline-treated group served as a negative control.

Prior to initiating transplantation, the BLM dose and evaluation time point were determined based on prior optimization guidance from supplier protocols and internal testing (Fig 3A). Morphological characterization of WKY-derived rPFs revealed that stimulation with TNF-α and IL-4 induced elongation of fibroblast morphology and upregulation of VCAM-1 expression, comparable to the response observed in human pulmonary fibroblasts under similar cytokine conditions (Fig 3B, 3C).

## Therapeutic effects of intravenously injected allogeneic pulmonary fibroblasts in fibrotic rat lungs

One day after intravenous transplantation of allogeneic pulmonary fibroblasts derived from WKY rats, cell engraftment in the lungs of recipient BN rats was assessed using the IVIS Spectrum imaging system (Perkin Elmer, SP-BFM-T1). The fluorescent signal confirmed that the injected cells localized exclusively to the lungs (Fig 4A).

Masson's trichrome staining was performed on lung sections collected at the experimental endpoint. On "Day 0" (i.e., 5 days after BLM administration, prior to cell or saline injection), histological evidence of early fibrosis—such as interstitial thickening and collagen deposition—was already apparent. In the control group (saline only), these fibrotic changes had markedly progressed, with architectural distortion, collapsed alveolar structures and dense extracellular matrix

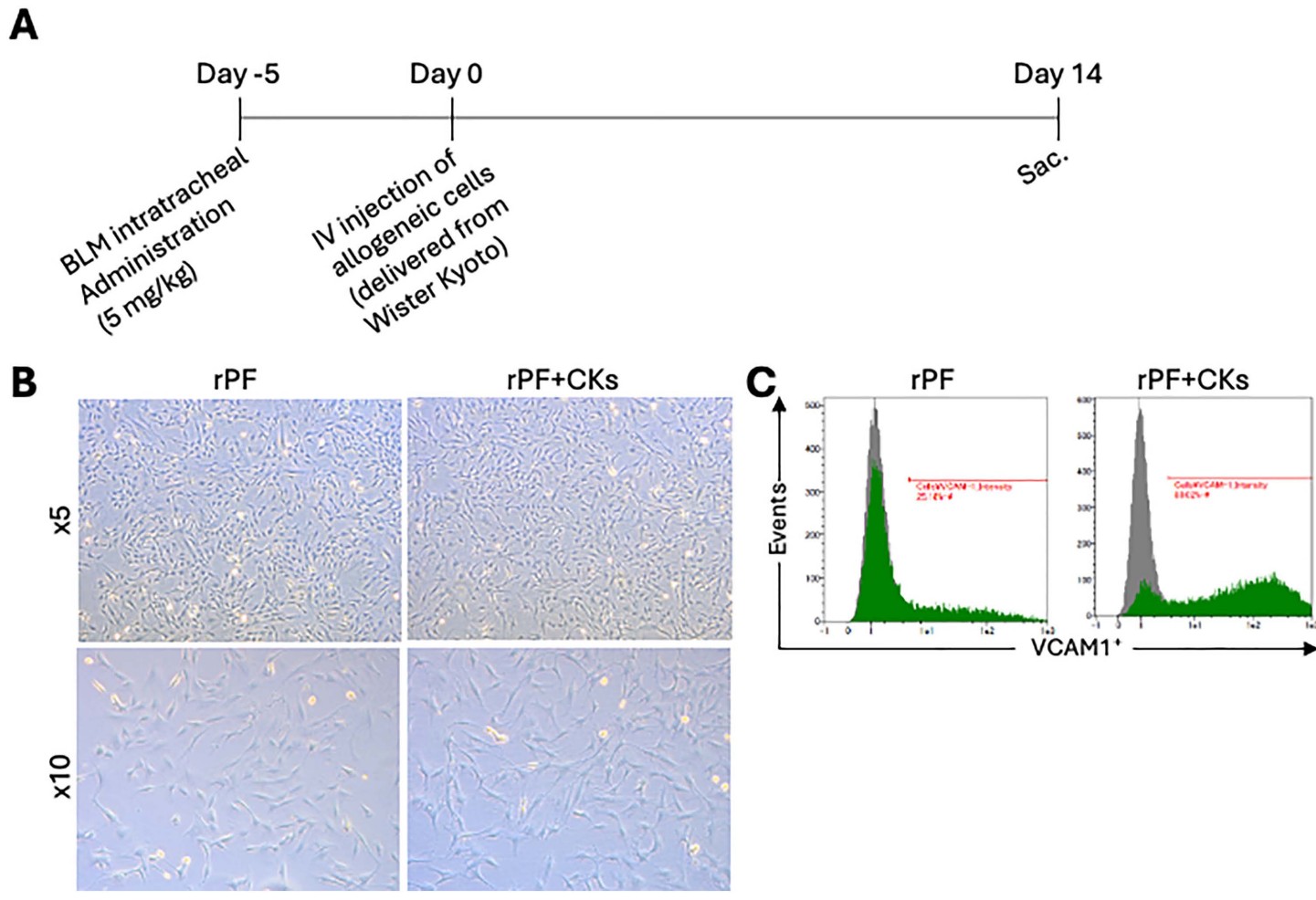

**Fig 3. Efficacy of allogeneic rPF+CKs in a rat model of pulmonary fibrosis. (A)** Experimental timeline of the allogeneic transplantation study. **(B)** Representative phase-contrast images of rPFs without stimulation (left) and rPFs stimulated with TNF-α and IL-4 (rPF+CKs, right), captured at ×5 (upper) and ×10 (lower) magnifications. Cytokine stimulation induced morphological elongation in rPFs, particularly evident at higher magnification. **(C)** Flow cytometry of VCAM-1 expression on the cell surface of rPFs and rPF+CKs.

accumulation. In contrast, lungs from rats treated with either rPF or rPF+CKs showed visibly reduced fibrotic lesions, with more preserved alveolar architecture and less collagen staining compared to both Day 0 and the control group (Fig 4B).

Quantitative analysis of fibrotic area, performed using a semi-automated ImageJ/Fiji-based method as previously described [12], showed a significant reduction in the rPF-treated group relative to the control, with an even greater—though not statistically significant—inhibitory trend observed in the rPF+CKs group (Fig 4C).

## Molecular and histological evidence of lymphangiogenic activation by allogeneic cytokine-stimulated fibroblasts

RT-qPCR analysis of lung tissue revealed that *Ccl2 (Mcp1),* a chemokine associated with poor prognosis in idiopathic pulmonary fibrosis [14], was significantly downregulated in the rPF+CKs-treated group.

Additionally, the expression levels of *IL6* (inflammatory cytokine), *αSMA* (myofibroblast marker), and *Col1a1* (fibrotic marker) showed reductions in both the fibroblast-treated groups, with rPF+CKs tending toward greater reductions than rPF, though these differences did not reach statistical significance (Fig 5A).

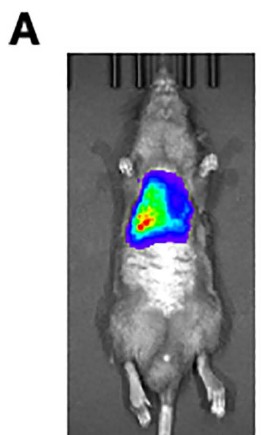

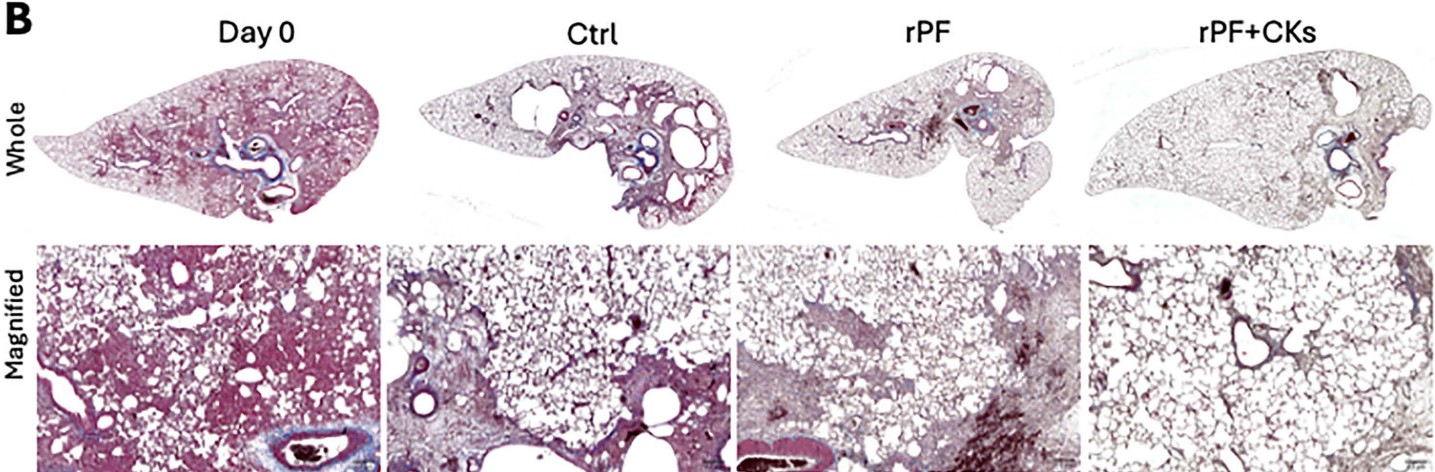

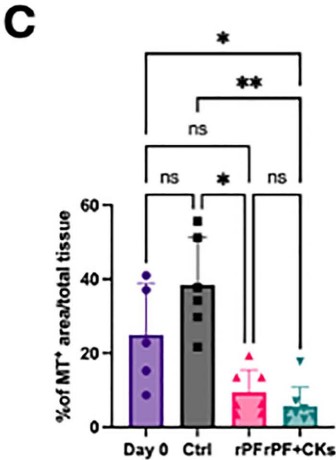

**Fig 4. Engraftment and therapeutic effects of pulmonary fibroblasts. (A)** IVIS imaging performed one day after cell administration showing pulmonary localization of transplanted fibroblasts. **(B)** Representative Masson trichrome-stained lung sections from each group (Day 0, saline (Ctrl), rPF, and rPF+CKs). For each group, whole lung images (top row) and corresponding magnified views (bottom row) are shown to highlight differences in fibrosis and tissue architecture. **(C)** Quantification of fibrotic area. Data are presented as mean±SEM. Statistical analysis was performed using the Kruskal–Wallis test, followed by Dunn's multiple comparisons test. *p<0.05, **p<0.01 vs. indicated groups.

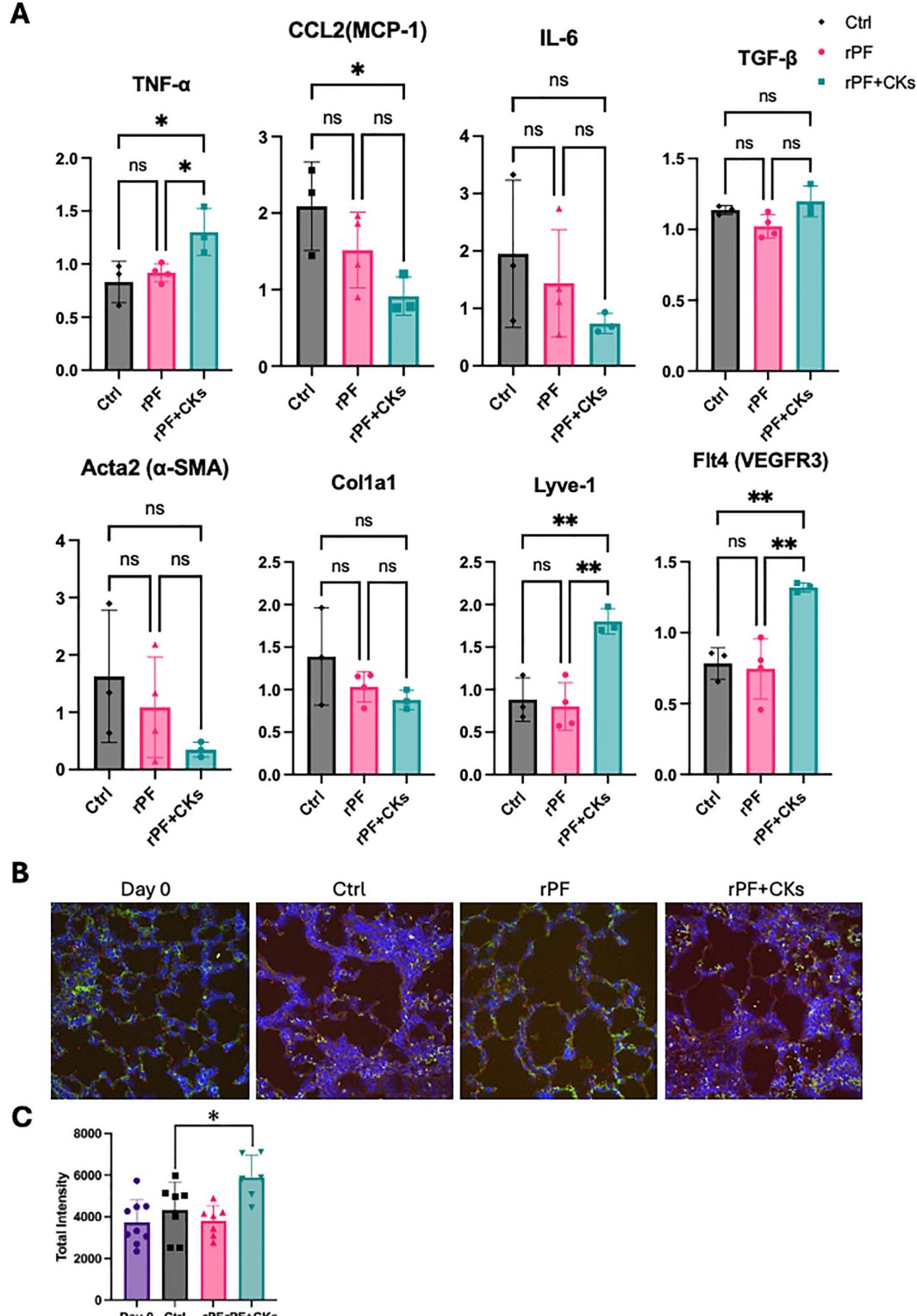

**Fig 5. Gene expression analysis and Lyve-1 localization in lung tissue. (A)** RT-qPCR analysis of fibrosis- and lymphangiogenesis-related genes in lung tissues. *P<0.05, **P<0.01 by one-way ANOVA followed by Tukey's multiple comparison test. **(B)** Representative double-staining images combining WGA immunofluorescence (green) and *Lyve1* ISH (red), with DAPI counterstain (blue). **(C)** Quantification of ISH signal (Total Intensity) in lung

sections following rPF and rPF + CKs treatment.Total fluorescence intensity per image was measured in each group (Day 0, rPF, rPF + CKs, and Ctrl). Sample sizes were as follows: Day 0 = 9 images, Ctrl = 7 images, rPF = 7 images, rPF + CKs = 6 images. Welch's t-test was performed to compare each group to the Ctrl. *p < 0.05 vs. Ctrl (Welch's t-test).

Conversely, lymphatic markers *Lyve1* and *Vegfr3* were significantly upregulated compared with the control and rPF groups, indicating robust activation of lymphangiogenic pathways by cytokine stimulation (Fig 5A).

To visualize the spatial distribution of this lymphangiogenic response, in situ hybridization (ISH) targeting *Lyve1* mRNA was performed. Red fluorescent dots indicating individual *Lyve1* transcripts were predominantly observed in the transitional zones between fibrotic and preserved alveolar regions. In addition to manual quantification, a complementary quantitative analysis was conducted using the total fluorescence intensity (Total Intensity) of ISH signals. In this analysis, only red dots co-localized with DAPI-stained nuclei were included, enabling selective quantification of cell-associated *Lyve1* transcripts. The cumulative fluorescence from these dots was calculated to estimate the overall transcript abundance. Since Total Intensity reflects both the number of expressing cells and their expression levels, this metric served to capture global changes in gene expression. Notably, the rPF + CKs group showed significantly higher Total Intensity than the control group (Fig 5B, 5C).

These findings suggest that pulmonary fibroblasts with lymphangiogenic capacity may serve as a promising therapeutic modality for pulmonary fibrosis, even in allogeneic transplantation settings.

### Plasma biomarkers support therapeutic efficacy of allogeneic pulmonary fibroblasts in bleomycin-induced lung injury

To evaluate the systemic condition associated with pulmonary fibrosis, plasma levels of surfactant protein D (SP-D) and fibrinogen were measured.

SP-D is a clinically established biomarker for interstitial lung diseases including idiopathic pulmonary fibrosis (IPF), reflecting the severity of the disease. Fibrinogen was selected based on previous reports indicating an inverse correlation between its plasma concentration and pulmonary function parameters such as %FEV1, suggesting its utility as a surrogate marker for lung function decline [15].

SP-D levels were significantly reduced in both the control and fibroblast-treated groups compared to Day 0 (five days after BLM administration). However, the magnitude of reduction was greater in the cell-treated groups, suggesting a potential improvement in lung injury status (Fig 6A).

Fibrinogen levels were significantly reduced in both the saline control and cell-treated groups compared to Day 0 (five days after BLM administration), as determined by one-way ANOVA followed by Dunnett's multiple comparison test. Although no significant differences were observed between the saline control and cell-treated groups, a trend toward lower SP-D levels—particularly in the rPF + CKs group—suggests a potentially enhanced therapeutic effect (Fig 6B).

These results suggest that systemic improvement, as reflected by plasma biomarker profiles, was more evident in the fibroblast-treated groups—particularly in the rPF + CKs group—supporting the enhanced therapeutic potential of cytokine-stimulated pulmonary fibroblasts.

## Discussion

In our previous studies, we demonstrated that cardiac fibroblasts derived from both HFrEF and HFpEF models could acquire lymphangiogenic properties when stimulated with cytokines such as TNF-α and IL-4 [13]. Similarly, Baluk et al. reported that expansion of the lymphatic network attenuated macrophage accumulation and fibrosis in murine models of pulmonary fibrosis [16]. These findings suggest that promotion of lymphangiogenesis may facilitate the clearance of excess extracellular matrix components, immune cells, and interstitial fluid, thereby contributing to fibrosis resolution across different organs.

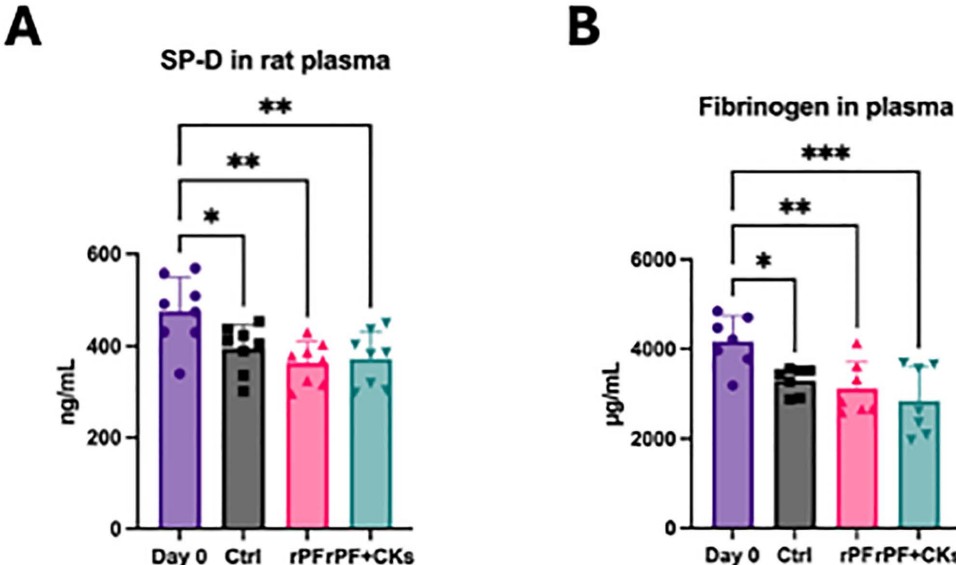

**Fig 6. Plasma biomarkers of pulmonary fibrosis. (A)** Effect of allogeneic rPF or rPF+CKs cell treatment on plasma levels of surfactant protein D (SP-D) levels. **(B)** Effect of allogeneic rPF or rPF+CKs cell treatment on plasma fibrinogen levels.Pulmonary fibrosis was induced in mice (Day 0), followed by treatment with saline (Ctrl), allogeneic pulmonary fibroblasts (rPF), or cytokine-enriched fibroblasts (rPF+CKs).Plasma samples were collected on day 14 and analyzed for fibrinogen concentration using ELISA.Sample sizes (animal numbers) were as follows: SP-D: Day 0=8, Ctrl=8, rPF=8, rPF+CKs=8 Fibrinogen: Day 0=7, Ctrl=7, rPF=7, rPF+CKs=7.

In this study, we demonstrated that pulmonary fibroblasts, when stimulated with TNF-α and IL-4, acquire similar lymph-angiogenic phenotypes, characterized by increased *ADM* and *VEGFC* expression and enhanced tube formation capacity in vitro (Figs 1). These findings indicate that fibroblasts in different tissues may possess plasticity to adopt reparative lymphangiogenic functions under specific cytokine environments.

Lymphangiogenesis has historically been underappreciated in the context of fibrosis; however, growing evidence indicates that insufficient lymphatic drainage contributes to the persistence of fibrotic tissue by impairing interstitial fluid clearance, delaying inflammatory resolution, and perpetuating myofibroblast activation [17]. *ADM* and *VEGFC*, both upregulated in our cytokine-stimulated fibroblasts, are well-known lymphangiogenic mediators involved in vascular remodeling and interstitial homeostasis [13]. In pulmonary fibrosis, where chronic alveolar epithelial injury leads to excessive ECM accumulation, enhancing lymphatic clearance pathways may represent a complementary antifibrotic mechanism [18].

One of the advantages of this approach is its potential applicability in allogeneic settings. Autologous cell therapies are limited by manufacturing time and patient deterioration during production [19]. In contrast, allogeneic pulmonary fibroblasts demonstrated low immunogenicity in this study, characterized by absent expression of co-stimulatory molecules such as CD80 and CD86 (Figs 2). This immunological profile resembles that of mesenchymal stromal cells (MSCs), which are widely studied for allogeneic therapies. MSCs have been shown to express low levels of MHC I and no MHC II, CD80, or CD86 in vitro [20].

However, in vitro immunogenicity profiles do not fully predict in vivo immune responses. Allogeneic cell transplantation may elicit immune responses including allo-antibody formation, innate immune activation, and delayed rejection, even when in vitro immunophenotyping suggests low immunogenicity [21,22]. In this study, all animals survived the two-week observation period without clinical signs of acute rejection, though we did not perform comprehensive immunological monitoring. During this early period, the observed therapeutic effects are likely mediated predominantly through paracrine signaling prior to immune-mediated clearance rather than long-term engraftment [23]. Regarding tumorigenicity, no

evidence suggestive of malignant transformation was observed in the present short-term culture and observation period using primary cells (passages 3–8) without genetic modifications. However, we emphasize that dedicated long-term tumorigenicity and immunogenicity studies will be required in accordance with regulatory guidelines [24].

Whether the lymphangiogenic phenotype remains stable in the fibrotic microenvironment also needs consideration. While TGF-β-rich conditions can cause fibrogenic changes in some cell populations [25], our gene expression analyses showed decreased pro-fibrotic markers (*Col1a1, α-SMA*) and increased lymphangiogenesis-related genes (*Lyve1, Vegfr3*) in treated lungs (Figs 5), suggesting that the therapeutic phenotype was maintained. However, lineage tracing studies are needed for definitive characterization. The two-week observation period in this study does not allow assessment of long-term immune responses and sustained therapeutic effects. Future studies should include extended safety evaluations, comprehensive immunological profiling, and studies using cell labeling to track donor cell fate and phenotypic stability.

In our in vivo model, allogeneic administration of cytokine-stimulated pulmonary fibroblasts significantly reduced fibrotic area, decreased fibrosis-associated gene expression (*Col1a1, α-SMA*), and upregulated lymphangiogenesis-related markers (*Lyve1, Vegfr3*). Although the increase in *Lyve1* ISH signal did not reach statistical significance, consistent trends across multiple readouts suggest activation of lymphangiogenic pathways (Figs 4,5).

The observed increases in lymphatic marker expression (*Lyve1, Vegfr3*) and reduction in fibrosis cannot be attributed solely to direct differentiation or structural incorporation of transplanted PFs without considering host-mediated mechanisms. Lymphangiogenesis during fibrosis has been shown to involve multiple endogenous processes, including activation and expansion of resident lymphatic endothelial cells, as well as macrophage transdifferentiation into lymphatic endothelial cell progenitors [26,27]. In fibrotic microenvironments, both sprouting of existing lymphatic vessels and recruitment of bone marrow-derived cells contribute to lymphatic network expansion, with inflammatory cells such as macrophages playing key roles through VEGF-C secretion and direct cellular contributions [26,27]. In the absence of lineage tracing or cell fate-mapping approaches, it is not possible to definitively distinguish whether transplanted PFs directly acquire lymphatic endothelial-like characteristics or whether the observed responses are primarily mediated by host cells. Consistent with our previous work demonstrating paracrine mechanisms in cytokine-stimulated cardiac fibroblasts [13], and given the transient nature of allogeneic cell survival (as discussed above), we propose that paracrine modulation of the host microenvironment through secreted factors such as VEGF-C and ADM represents a plausible and likely mechanism in our pulmonary fibroblast system. However, we clearly acknowledge that this interpretation remains inferential, and that future studies incorporating lineage tracing or similar approaches will be required to directly assess the cellular origin and structural contribution to lymphatic remodeling.

Notably, our results demonstrated that the fibrotic area in fibroblast-treated lungs was not only reduced compared to the control group, but also significantly decreased relative to the baseline Day 0 group (Fig 4B, 4C). This finding suggests that the therapy may contribute not only to preventing further progression of fibrosis, but also to actively reversing pre-existing fibrotic tissue. Such an effect indicates the potential for regenerative remodeling rather than merely stabilization, representing a significant advancement over existing antifibrotic therapies that largely aim to halt disease progression without reversing tissue damage.

Several study limitations warrant consideration. While histological analyses demonstrated significant reductions in fibrotic area and RT-qPCR analyses showed statistically significant increases in lymphatic marker gene expression (*Lyve1, Vegfr3*) with rPF+CKs treatment, the corresponding *Lyve1* in situ hybridization signal increase did not reach statistical significance and should be interpreted as a trend rather than a definitive effect. Similarly, other parameters including fibrotic markers (*IL6, α-SMA, Col1a1*) demonstrated trends toward improvement without reaching statistical significance. Interpretation of plasma biomarkers (SP-D and fibrinogen) should also be tempered by the known variability and partial spontaneous resolution characteristic of the bleomycin model; observed reductions in these markers, while consistent with therapeutic benefit, should not be interpreted as definitive therapeutic effects without stronger statistical support. Additionally, our relatively short 2-week observation period and modest sample size may limit statistical power to detect

subtle differences between treatment groups. These limitations notwithstanding, the consistent patterns of improvement across multiple outcome measures suggest potential therapeutic benefit warranting further investigation.

Surfactant protein D (SP-D), which reflects alveolar epithelial injury and is widely used as a clinical biomarker in IPF, was significantly reduced in both the rPF and the rPF + CKs groups compared to the control, suggesting attenuation of lung injury. Previous studies have shown that serum [28,29], including predictive value for antifibrotic therapy efficacy [30,31]. Fibrinogen levels, which correlate with lung functional decline and reflect both systemic inflammation and pro-fibrotic remodeling, were also markedly decreased, particularly in the rPF + CKs group. Elevated plasma fibrinogen has been associated with enhanced coagulation cascade activation, microvascular injury, and extracellular matrix deposition in progressive fibrotic lung disease. Furthermore, fibrinogen elevation may contribute to impaired alveolar gas exchange by promoting vascular remodeling and increased blood viscosity, ultimately correlating with worsened respiratory function and prognosis. While some reduction was also observed in the control group (Figs 6), the more pronounced decreases in treated groups, particularly rPF + CKs, suggest active therapeutic effects. These results support the potential of fibroblast-based therapies to enhance the resolution of inflammation and fibrotic remodeling.

Interestingly, a modest reduction in both SP-D and fibrinogen levels was also observed in the saline-treated control group compared to Day 0 (Figs 6), which may reflect the natural resolution of acute inflammation following the peak injury phase induced by bleomycin. This trend has been previously reported in rodent models of pulmonary fibrosis, where SP-D levels tend to rise rapidly after BLM exposure and gradually decline thereafter as the inflammatory response subsides. However, the more pronounced and statistically significant reductions observed in the rPF and rPF + CKs groups suggest an active therapeutic effect beyond natural recovery, particularly in the cytokine-stimulated group. These results reinforce the potential of fibroblast-based therapies to enhance the resolution of inflammation and fibrotic remodeling.

Taken together, these results support the concept that modulation of lymphangiogenesis through cytokine-conditioned pulmonary fibroblasts may represent a novel antifibrotic strategy. Future studies will need to optimize dosing regimens, assess long-term safety and efficacy, and validate these findings in large animal models and human cells. Additionally, further mechanistic studies are warranted to clarify how ADM and VEGF-C coordinate fibroblast-lymphatic crosstalk and contribute to ECM remodeling. Ultimately, this lymphangiogenesis-focused approach may offer a complementary or alternative pathway to current antifibrotic therapies for pulmonary fibrosis.

## Supporting information

**S1 Table. Antibodies and reagents used for FACS and immunohistochemistry.**
(PDF)

**S2 Table. Primers and Probes used for RT-qPCR.**
(PDF)

## Acknowledgments

The authors would like to thank The Jackson Laboratory Japan, Inc. for their assistance with the bleomycin model in rats.

Furthermore, the authors wish to acknowledge Division for Medical Research Engineering, Nagoya University Graduate School of Medicine for usage of SpinSR10 (Evident, Tokyo, Japan).

## Author contributions

**Conceptualization:** Yuimi Matsuoka, Takahiro Iwamiya.

**Data curation:** Yuimi Matsuoka.

**Investigation:** Yuimi Matsuoka, Makoto Matsuyama.

**Methodology:** Yuimi Matsuoka.

**Project administration:** Yuuki Shimizu, Koji Sakamoto.

**Supervision:** Toyoaki Murohara, Takahiro Iwamiya.

**Writing – original draft:** Yuimi Matsuoka.

**Writing – review & editing:** Yuimi Matsuoka, Yuuki Shimizu, Koji Sakamoto, Takahiro Iwamiya.

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
