## [Decision Letter · Decision Letter 0]

11 Dec 2025

Dear Dr. Matsuoka,

We look forward to receiving your revised manuscript.

Kind regards,

Ramada Rateb Khasawneh

Academic Editor

PLOS One

Journal Requirements:

2. Thank you for your submission to PLOS One. We note that your study design may include death of a regulated animal as a likely outcome or planned experimental endpoint. At this time, we request that you please report additional details in your Methods section regarding animal care and use for the survival study, as per our editorial guidelines (http://journals.plos.org/plosone/s/submission-guidelines#loc-humane-endpoints).

For easy reference, we have attached a checklist that may be relevant for your submission. Please complete all items on the checklist at the following link: http://journals.plos.org/plosone/s/file?id=bb1d/plos-one-humane-endpoints-checklist.docx

Please upload the completed checklist as file type “Other” when resubmitting your manuscript. This document is for internal journal use only and will not be published if your article is accepted. We very much appreciate your attention to these requests and support of improved reporting standards in PLOS One submissions.

To comply with PLOS One submissions requirements, in your Methods section, please provide additional information regarding the experiments involving animals and ensure you have included details on (1) methods of sacrifice, (2) methods of anesthesia and/or analgesia, and (3) efforts to alleviate suffering.

“I have read the journal’s policy and the authors of this manuscript have the following competing interests: YM is affiliated with LYMPHOGENiX Inc., which collaborated with Nagoya University in the execution of this study. This affiliation did not influence the study design, data collection and analysis, decision to publish, or preparation of the manuscript.”

We note that one or more of the authors are employed by a commercial company: LYMPHOGENiX Ltd

2) Please also provide an updated Competing Interests Statement declaring this commercial affiliation along with any other relevant declarations relating to employment, consultancy, patents, products in development, or marketed products, etc.

Within your Competing Interests Statement, please confirm that this commercial affiliation does not alter your adherence to all PLOS ONE policies on sharing data and materials by including the following statement: ""This does not alter our adherence to  PLOS ONE policies on sharing data and materials.” (as detailed online in our guide for authors http://journals.plos.org/plosone/s/competing-interests) . If this adherence statement is not accurate and  there are restrictions on sharing of data and/or materials, please state these. Please note that we cannot proceed with consideration of your article until this information has been declared.

**Additional Editor Comments:**

Overall, this is an interesting and ambitious study with a promising conceptual framework. The approach has clear potential, but several areas would benefit from greater clarity, contextualization within existing literature, and a more cautious interpretation of the findings. Below, I outline the major strengths, followed by key concerns and recommendations for improvement.

1. Immunogenicity and Allogeneic Use — Need for Further Characterization

The manuscript reports “minimal immunogenicity” of cytokine-stimulated PFs based on the absence of HLA-II and co-stimulatory molecules (CD80/86). While this aligns with characteristics observed in MSC-based therapies, such in vitro findings do not fully predict in vivo immune responses. Allogeneic cell transplantation can still lead to immunological events such as allo-antibody formation, innate immune activation, or delayed rejection. A more comprehensive discussion of these potential risks—along with considerations regarding long-term engraftment, persistence, and possible tumorigenicity—would substantially strengthen the manuscript.

Furthermore, clarification is needed regarding the long-term stability and fate of transplanted PFs. It remains unclear whether cytokine-stimulated PFs sustain a lymphangiogenic phenotype over time or undergo phenotypic drift, dedifferentiation, or aberrant contributions to tissue remodeling after transplantation.

2. Specificity of the Observed Effects — Role of PFs vs. Other Cell Populations

The observed increases in lymphatic marker expression (LYVE-1, VEGFR3) and the reduction in fibrosis are attributed to the transplanted PFs. However, existing literature indicates that lymphangiogenesis during fibrosis often involves endogenous mechanisms, including activation of resident lymphatic endothelial cells or macrophage-derived lymphatic-like cells. Alternative explanations—such as host-mediated lymphatic remodeling or macrophage transdifferentiation—should be acknowledged and discussed.

If feasible, lineage tracing or similar techniques would provide stronger evidence regarding whether transplanted PFs directly differentiate into lymphatic endothelial-like cells or contribute structurally to lymphatic remodeling, rather than acting solely through paracrine signaling.

3. Quantitative Strength of Findings and Statistical Interpretation

Several results are described as demonstrating a “greater inhibitory trend,” particularly in the rPF+CKs group compared to rPF alone; however, these differences were not statistically significant. Such findings should be presented explicitly as trends and not interpreted as definitive effects. Similarly, although increases in lymphatic markers and ISH signals are reported, the manuscript acknowledges that the increase in LYVE-1 ISH did not reach statistical significance. These limitations should be more transparently addressed, ideally in both the Results and Discussion sections.

Interpretation of plasma biomarker trends (SP-D, fibrinogen) should also be tempered by the known variability of the bleomycin model and its partial spontaneous resolution. Overstating these findings as clear therapeutic effects may be misleading without stronger statistical support.

Reviewers' comments:

Reviewer's Responses to Questions

**Comments to the Author**

1. Is the manuscript technically sound, and do the data support the conclusions?

Reviewer #1: Yes

Reviewer #2: Yes

2. Has the statistical analysis been performed appropriately and rigorously?

Reviewer #1: Yes

Reviewer #2: Yes

3. Have the authors made all data underlying the findings in their manuscript fully available?

Reviewer #1: Yes

Reviewer #2: Yes

4. Is the manuscript presented in an intelligible fashion and written in standard English?

Reviewer #1: Yes

Reviewer #2: Yes

Reviewer #1: Excellent manuscript regarding Cytokine-stimulated pulmonary fibrobsis. Pulmonary fibroblastsmay serve as a novel cell

source for antifibrotic therapy by modulating lymphangiogenesis and tissue

remodeling, providing a potential alternative to conventional stem cell-based

approaches for fibrotic lung diseases as stated by the authors. This may be a new molecular pathway for potential treatment targets in pulmonary hypertension.

Reviewer #2: I would like to thank you and your colleagues for this excellent scientific work and for your valuable contribution to the advancement of science.

Review letter has been uploaded seperately.

Thank you.

**Do you want your identity to be public for this peer review?** For information about this choice, including consent withdrawal, please see our Privacy Policy

Reviewer #1: **Yes:** Afendoulis Dimitrios

Reviewer #2: No

---

## [Author Response · Author response to Decision Letter 1]

12 Jan 2026

We thank the Academic Editor and reviewers for their constructive feedback. We have carefully addressed all comments raised, as detailed in the attached "Response to Reviewers" document. Major revisions include:

- Expanded Discussion to address immunogenicity, phenotypic stability, and alternative mechanisms

- Revised Results and Discussion to present non-significant findings as trends

- Extensively expanded Animal experiments section with humane endpoints details

- Updated Funding and Competing Interests statements

Please see the attached Response to Reviewers document for detailed point-by-point responses.

---

## [Decision Letter · Decision Letter 1]

26 Jan 2026

Pulmonary fibroblasts activated by the addition of TNF-α and IL-4 enhance lymphangiogenic capacity and ameliorate lung fibrosis in an allogeneic rat model

PONE-D-25-47407R1

Dear Dr. Matsuoka,

We’re pleased to inform you that your manuscript has been judged scientifically suitable for publication and will be formally accepted for publication once it meets all outstanding technical requirements.

Kind regards,

Ramada Rateb Khasawneh

Academic Editor

PLOS One

Additional Editor Comments (optional):

Good Luck

Reviewers' comments:

Reviewer's Responses to Questions

**Comments to the Author**

Reviewer #1: All comments have been addressed

Reviewer #2: All comments have been addressed

2. Is the manuscript technically sound, and do the data support the conclusions?

Reviewer #1: Yes

Reviewer #2: Yes

3. Has the statistical analysis been performed appropriately and rigorously?

Reviewer #1: Yes

Reviewer #2: Yes

4. Have the authors made all data underlying the findings in their manuscript fully available?

Reviewer #1: Yes

Reviewer #2: Yes

5. Is the manuscript presented in an intelligible fashion and written in standard English?

Reviewer #1: Yes

Reviewer #2: Yes

Reviewer #1: Interesting manuscript regarding trial of cytokine conditioned fibroblasts, which may be a new pathway

For treatment of pulmonary fibrosis. Most of the reviewers comments have been addressed, enforcing the impact of the data

Presented in the manuscript. My recommendation is to proceed for publication.

Reviewer #2: I would like to thank all the authors for this excellent research. I believe that this study will make a significant contribution to the existing literature and serve as a valuable reference for future research.

**Do you want your identity to be public for this peer review?** For information about this choice, including consent withdrawal, please see our Privacy Policy

Reviewer #1: **Yes:** Afendoulis Dimitrios

Reviewer #2: No

---

## [Editor Report · Acceptance letter]

PONE-D-25-47407R1

PLOS One

Dear Dr. Matsuoka,

I'm pleased to inform you that your manuscript has been deemed suitable for publication in PLOS One. Congratulations! Your manuscript is now being handed over to our production team.

Kind regards,

on behalf of

Dr. Ramada Rateb Khasawneh

Academic Editor

PLOS One